# An Automatic Jujube Fruit Detection and Ripeness Inspection Method in the Natural Environment

**Defang Xu** [1], **Huamin Zhao** [2,*], **Olarewaju Mubashiru Lawal** [3], **Xinyuan Lu** [2], **Rui Ren** [2] and **Shujuan Zhang** [2]

1   Department of Economic and Management, Lvliang University, Lvliang 033001, China
2   College of Agricultural Engineering, Shanxi Agricultural University, Jinzhong 030801, China
3   Sanjiang Institute of Artificial Intelligence & Robotics, Yibin University, Yibin 644000, China
*   Correspondence: zhaohuamin@sxau.edu.cn

**Abstract:** The ripeness phases of jujube fruits are one factor mitigating against fruit detection, in addition to uneven environmental conditions such as illumination variation, leaf occlusion, overlapping fruits, colors or brightness, similar plant appearance to the background, and so on. Therefore, a method called YOLO-Jujube was proposed to solve these problems. With the incorporation of the networks of Stem, RCC, Maxpool, CBS, SPPF, C3, PANet, and CIoU loss, YOLO-Jujube was able to detect jujube fruit automatically for ripeness inspection. Having recorded params of 5.2 m, GFLOPs of 11.7, AP of 88.8%, and a speed of 245 fps for detection performance, including the sorting and counting process combined, YOLO-Jujube outperformed the network of YOLOv3-tiny, YOLOv4-tiny, YOLOv5s, and YOLOv7-tiny. YOLO-Jujube is robust and applicable to meet the goal of a computer vision-based understanding of images and videos.

**Keywords:** YOLO-jujube; fruit detection; ripeness inspection; sorting and counting





## 1. Introduction

Jujube (*Ziziphus jujuba*) is a fruit native to China, which is referred to as Chinese date or red date in a dried state. It is high in vitamins and minerals, and low in calories. The shape of jujube fruit ranges from round to pear-shaped with a thin, edible skin and whitish flesh. The fruit turns from green to yellow-green with red-brown spots when ripe, and later wrinkles, softens, and becomes fully red. The redder the fruit, the sweeter the fruit is. At the same time, the mature phases of the fruit can be divided into white, crisp, and fully ripe using color, flesh firmness, and composition (starch, sugar, acid, water). White mature is close to full size and shape, which changes from green to greenish white with less juice and sugar and more starch; at this stage, the fruit is picked for candying. Crisp mature is half to fully red in color; the fruit becomes crisp, juicy, and sweet, and is harvested for fresh consumption and a prolonged shelf-life. Fully mature fruit changes to full dark red and becomes wrinkled with an increased sugar and decreased water content; at this stage, it is mainly used for dried state fruit. For these reasons, the ripeness inspection of the fruit for sorting becomes a difficult task, and hand-picking of the mature fruit is tedious and time consuming. Consequently, it is necessary to develop an automatic jujube fruit detection method for ripeness inspection. This is to replace the traditional hand-picking method of fruit monitoring for quick and accurate sorting and counting.

Fruit detection using deep learning is a computer vision method used to localize and classify targets in an image or video. It has been widely utilized for ripeness recognition, yield prediction [1], harvesting/picking robot applications [2–4], fruit-quality detection [5], fruit estimation, fruit counting [6], etc. Notwithstanding, the complex and changeable natural environment can be challenging for fruit detection. Leaf occlusion, overlapping fruits, variation of illumination, colors or brightness, similar plant appearance as the background, and nonstructural fields [7], among others, are some of these factors generally experienced in fruits detection. Additionally, the intelligent perception and dataset acquisition of jujube

fruits for sorting and counting due to their ripeness stages, clusters growth, and complex background of leaves, branches, and stems becomes one of the most difficult tasks for the fruit detection. Interestingly, the fruit detection model is a vital aspect of an automatic recognition system, which contains a computer vision and computational platform; in other words, the success of the detection, sorting, and counting of target fruit in an image or video depends on the robustness of the fruit detection model.

You Only Look Once (YOLO) is a single-stage object detector that has shown remarkable performance for fruit detection. The quest for speed, low computational cost, and applicability to low power computing devices lead to the introduction of the tiny YOLO structure. Baes on the YOLOv3-tiny [8], a fast and accurate kiwifruit detection method developed by Fuet al. [9] reported an average precision (AP) of 90.05% and inference time of 29.4 frames per second (fps). However, the model weight-size is still large with a slower detection speed. The FL-YOLOv3-tiny proposed by Tan et al. [10] for underwater object detection noted an AP of 10.9% and a 29% fps improvement on YOLOv3-tiny. Nevertheless, the robustness of the proposed model for generalization is still questionable. Gai et al. [11] reported an AP of 95.56% and a speed of 35.5 fps, which was an improvement on YOLOv3-tiny; however, it was not tested on fruits with complex background and it has a large weight-size with a slower detection speed. Bochkovskiy et al. [12] upgraded the YOLOv3 to YOLOv4. Similarly, Lawal [2] and Liu et al. [13] demonstrated that variation of illumination and occlusion factors of fruit detection is solvable using an improved YOLOv3 model. The YOLO-Oleifera model proposed by Tang et al. [14], based on modified YOLOv4-tiny, achieved an AP of 92.07% and had a weight-size of 29 MB and average detection time of 32.3 fps to detect each fruit image. However, the detection time is still slower with a large weight-size. A robust real-time pear fruit counter for mobile applications using YOLOv4-tiny by Parico et al. [15] recorded a speed of more than 50 fps and an AP of 94.19%; however, it had an associated weight-size of 22.97 MB, which defines a high computational cost. The GCS-YOLOv4-tiny [16] model for multi-stage fruit detection achieved an AP of 93.42%, with a model weight-size of 20.70 MB. Nevertheless, the model was not tested for real-time detection, missed the detection in dense small images, and the weight-size remained large. YOLOv5 [17], which has an excellent fast detection speed relative to YOLOv4, was presented. The apple target detection model based on the modified YOLOv5 developed by Yan et al. [18] reported an accuracy of 86.75% and a detection time of 66.7 fps; however, it still needs further detection time improvement. Owing to the version of YOLOv5s, Zhang et al. [19] introduced a ghost (Han et al. [20]) module and SCYLLA-IoU (SIoU) loss function into the network for improvement in the detection of dragon fruit in the natural environment, which had an AP of 97.4% with a weight-size of 11.5 MB. However, in this case, the target is obviously different from the background; therefore, the generalization of the model is yet to be ascertained on different fruit datasets with complex backgrounds, particularly for jujube fruits, and the weight-size remains large. The YOLOv5s-cherry, proposed by Gai et al. [21] for cherry detection, had a $F_1$ of 0.08 and 0.03, which were higher than the YOLOv4 and YOLOv5s, respectively, but also needs to be improved. A counting method of red jujube based on the modified YOLOv5s proposed by Qiao et al. [22] recorded an AP of 94% and a speed of 35.5 fps using a ShuffleNetv2 [23] backbone. Nevertheless, the robustness of the model is not certain because it was tested for only fully mature red jujube fruits and its detection speed is slower. Recently, the YOLOv7, introduced by Wang et al. [24], was reported to have surpassed other known object detectors, including YOLOv4 and YOLOv5, achieving the highest accuracy of 56.8% and detection speed in the range from 5 to 160 fps on the MS coco dataset. The extended efficient layer aggregate networks (E-ELAN) utilized by YOLOv7 primarily focus on the parameters and computational density of the model for performance improvement. Apart from Zhang et al. [19], who experimented on YOLOv7 and YOLOv7-tiny for dragon fruit detection and Chen et al. [25] with modified YOLOv7 for citrus detection, the investigation on jujube fruit detection using YOLOv7-tiny is seldom. A large weight-size and the parameters common to all mentioned YOLO variants are a

big challenge to realize a faster detection speed and deployment in low-power computing devices. More so, these methods were aimed at relatively sparse fruits to justify their high AP and cannot solve the detection problem of small and dense fruits having a similar image background. Therefore, it is necessary to address the factors of the fruit detection model, including large weight-size and parameters, low detection speed, and accuracy, and to investigate the seldom architecture using a jujube fruit dataset of small and dense images.

This study designed a detection network based on YOLOv5 to reduce the size of the model and improve the speed and accuracy, which was used for real-time detection, sorting, and counting of jujube fruit in complex scenes. The main contributions of this paper are as follows:

(1) To produce a robust jujube fruit image dataset including both white and crisp mature levels of ripeness with associated complex and changeable natural environments.
(2) Introduce the stem network for feature expression capability and the RCC network for increased precision learning in the YOLO-Jujube network.
(3) Develop a YOLO-Jujube method that is fast, accurate, smaller in parameters, and robust for generalization.
(4) Compare the performance of the proposed YOLO-Jujube with YOLOv3-tiny, YOLOv4-tiny, YOLOv5s, and YOLOv7-tiny methods.

## 2. Materials and Methods

### 2.1. Dataset Information

The jujube fruit images were collected from the fruit planting base in Gaolang Red Date Picking Garden, Linxian County, Luliang, Shanxi Province. Intelligent mobile phones (Huawei mate30pro and mate40pro) were used to collect the images of jujube fruits. At the white and crisp mature stages, the images were obtained in the morning, mid-day, and afternoon with a constantly changing shooting angle and distance. The captured images include complex and changeable conditions, such as varying maturity, leaf occlusion, fruit overlap occlusion, superimposed fruit, dense target fruit, branch occlusion, back light, front light, side light, and other fruit scenes. Figure 1 shows some of the obtained images and Table 1 provides the image dataset information under different conditions. A total of 1959 images were collected using a resolution of $1904 \times 4096$ and $2736 \times 3648$ pixels in JPG format, and recorded videos of jujube fruit in mp4 format (246.27 MB, 199.2 MB and 20.79 MB) were obtained to investigate the robustness and real-time detection speed. The images were randomly divided into the training set (80%), validation set (15%), and the test set (5%) for network training paradigms. The ground truth bounding boxes of each target in an image were manually hand labeled into two-classes: 80R (Crisp mature) and 70R (White mature) using the LabelImg graphical annotation tool. The assumed shape of the target fruit was drawn without considering the complicated condition of the image, and the annotation files were saved in the YOLO text format. The YOLO text format includes target class, coordinates, height, and width. Finally, a total of 10912, 1921, and 637 bounding boxes were generated from 1569 training set images, 292 validation set images, and 98 test set images,. Table 2 describes the annotation details of the images, but the videos were not annotated as unseen data to enable open counting of detection targets.

**Table 1.** Image dataset information under different conditions.

| Time of Day | Occlusions | Dense Fruits | Close Fruits | Light Variation | Others |
|:-----------:|:----------:|:------------:|:------------:|:---------------:|:------:|
| Morning     | 280        | 95           | 131          | 105             | 39     |
| Midday      | 268        | 99           | 150          | 110             | 33     |
| Afternoon   | 270        | 85           | 166          | 92              | 36     |

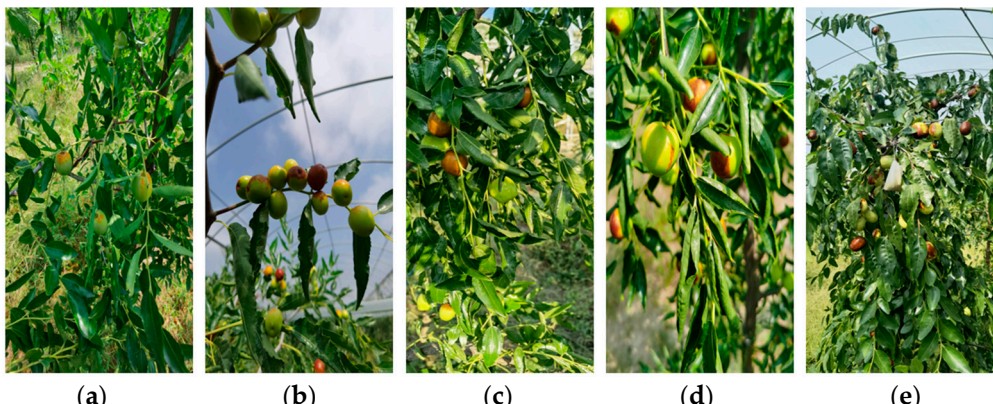

|           (a)            |            (b)            |            (c)            |            (d)            |            (e)            |

**Figure 1.** The conditions of images in the dataset under (**a**) similar background; (**b**) clusters; (**c**) occlusion; (**d**) front light; (**e**) dense target.

**Table 2.** Annotation count ($A_C$) details.

| Division | Images | 80R | 70R | Boxes |
|----------|--------|------|------|--------|
| Train | 1569 | 8460 | 2452 | 10,912 |
| Valid | 292 | 1487 | 434 | 1921 |
| Test | 98 | 494 | 143 | 637 |
| Total | 1959 | 10,441 | 3029 | 13,470 |

### *2.2. YOLO-Jujube*

As a build on the YOLOv5 [17], the proposed YOLO-Jujube method has a 1.0 depth and 1.0 width multiple, and consists of a backbone, neck, and head network, as shown in Figure 2. Figure 3 depicts all the components of the network. The backbone, termed the JujubeNet, integrates the Stem, RCC, Maxpool, CBS, and SPPF network modules for feature extraction. Stem is the first spatial down-sampling network used on the input image; it increases the generalization capability and reduces the computation complexity of the method. The incorporated RCC network combines ResNet [26] and CBS; both networks were directly concatenated for information sharing, where the complementary features of low-layer in CBS concatenates a high-layer in ResNet, as indicated in Equation (1), where $X \in R^{H \times W \times C1}$ is the low-layer features in CBS, C1 is the number of channels, and W and H is the width and height, respectively. $Y \in R^{H \times W \times C2}$ is the high-level features in ResNet. $O \in R^{H \times W \times (C1+C2)}$ is the concatenated features. The idea of the RCC is to enhance the capacity of the network to learn more diverse features by expanding the number of channels. The ResNet in RCC network is adopted to solve drops off from saturated accuracy for deeper neural network and is also responsible for reducing parameters and complexity. Meanwhile, the CBS is a convolution layer activated with SiLU [27] after the batch normalization (BN) layer, and Maxpool is basically for down-sampling. The added SPPF is a feature enhancement module that helps to reduce missed target detection. According to Jocher et al. [17], SPPF is faster than spatial pyramid pooling (SPP) [28].

$$O = [X, Y] \tag{1}$$

The neck part of YOLO-Jujube uses a path aggregation network (PANet) [29] for multi-scale feature fusion. The three feature layers of different scales obtained in the backbone continue to extract features in the neck network. PANet is used to promote a well-generalized model and to maintain a balance between accuracy and speed. With all things being the same as YOLOv5 for the neck, the number of C3 networks in Figure 2 was pruned to one to reduce the model size and foster the detection speed. The head network consumes map features from the neck to output the small, medium, and large scale of fruit detection. It uses bounding boxes (anchors) on these features for the purpose of producing

the fruit target class, score, and position. Similarly, non-maximum suppression (NMS) was applied to select the appropriate fruit target and remove redundant information, and complete intersection-over-union (CIoU) loss [30] function utilized for convergence speed and localization accuracy. CIoU loss considered overlapping areas, the center distance, and the aspect ratio of boxes to improve fruit detection accuracy.

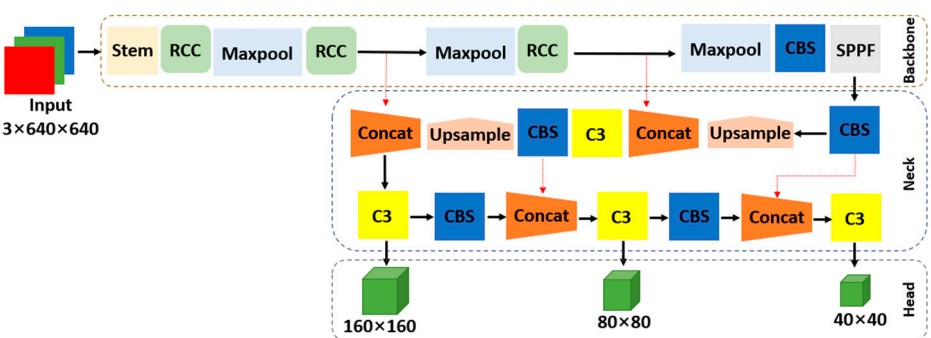

**Figure 2.** The entire network structure of the YOLO-Jujube method.

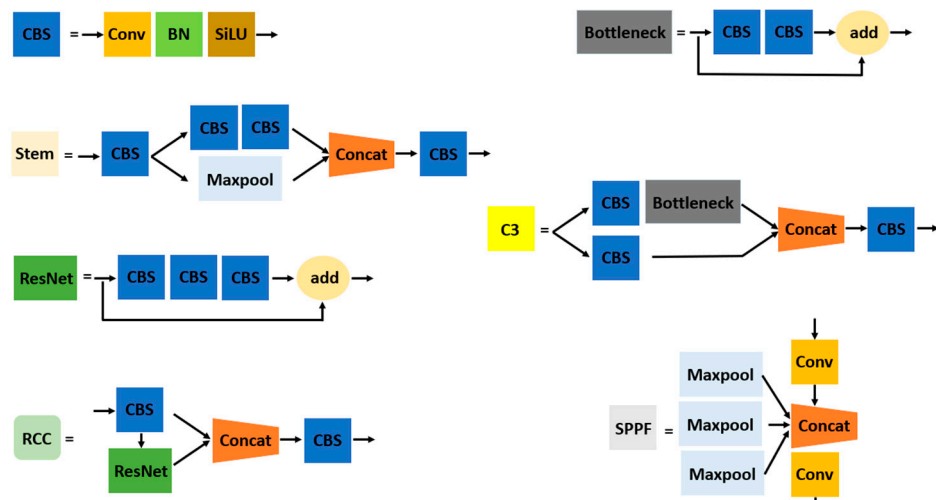

**Figure 3.** The component of the YOLO-Jujube method.

### 2.3. Experiment

The training and testing sets of this experiment were deployed on an Intel Core i7-12700 CPU @ 64-bit 4.90 GHz, 32 GB RAM, NVIDIA GeForce RTX 3060 GPU, python 3.19.13, and torch-1.11.0 + cu113 with Ubuntu22.04LTS operating system. The proposed YOLO-Jujube, including other compared methods as indicated in Table 3, takes an input of $640 \times 640$ pixels, 16 batch size, 0.937 momentum, 0.0005 weight decay, 0.2 IoU, 0.015 hue, 0.7 saturation, 0.4 lightness, 1.0 mosaic, and 120 epochs for training. The random initialization technique was adopted to initialize the weights for training all the methods from scratch.

**Table 3.** List of trained models.

| Methods | Backbone | Neck | Loss |
|---|---|---|---|
| YOLOv3-tiny | DarkNet | FPN | CIoU |
| YOLOv4-tiny | CSPDarkNet | PANet | CIoU |
| YOLOv5s | C3-CSPNet | PANet | CIoU |
| YOLOv7-tiny | E-ELAN | PANet-FPN | CIoU |
| YOLO-Jujube | JujubeNet | PANet | CIoU |

*2.4. Evaluation*

The evaluation metrics used for the jujube fruit detection performance were precision (P), recall (R), $F_1$, average precision (AP), speed, layers, number of parameters (Params), giga floating point operations per second (GFLOPs), and weight size. The P, R, $F_1$, AP, and speed can be measured using Equations (2)–(6), respectively. TP is the true positive (correct detections), FN is the false negative (missed detections), FP is the false positive (incorrect detections), and $P_{(R)}$ is P is a function of R. $F_1$ is the trade-off between P and R, and AP is the overall detection performance. The network tends to perform better with an increase in AP. Speed is measured in frames per second (fps). Params is used to measure the network complexity. The layer is a network topology. GFLOPs is the speed of the network. Size is used to measure the network weight.

$$P = \frac{TP}{TP + FP} \tag{2}$$

$$R = \frac{TP}{TP + FN} \tag{3}$$

$$F_1 = \frac{2 \times R \times P}{R + P} \tag{4}$$

$$AP = \int_0^1 P_{(R)} dR \tag{5}$$

$$Speed = 1/Inference \tag{6}$$

The percentage difference (d%) in Equation (7) is basically used when the direction of change is not known. It is measured as the absolute difference between two numbers: $|d_C\_A_C|$ divided by the average of those two numbers, where $d_C$ is the detection count while $A_C$ is the annotation count.

$$d\% = \frac{|d_C - A_C|}{(d_C + A_C)/2} \times 100 \tag{7}$$

## 3. Results and Discussion

*3.1. Accuracy and Speed*

The presented validation loss of box in Figure 4a is used to predict the performance of the methods by showing the consistent decreasing pattern. It measures the real location of the target fruit in an image. Owing to the level of a deeper neural network, the box validation loss of the YOLO-Jujube method is lower than other compared methods. This phenomenon confirms that when methods learn, performance improves. This means that a decreasing loss of the validation box is an increase in the AP, as shown in Figure 4. The results obtained on the valid-set in Figure 4b verifies that the AP level seen in YOLO-Jujube is higher than in YOLOv5s, YOLOv7-tiny, YOLOv4-tiny, and YOLOv3-tiny. At the same time, the tested detection speed shows that YOLOv3-tiny at 435 fps is more than YOLOv4-tiny at 313 fps, followed by YOLO-Jujube at 306 fps, YOLOv7-tiny at 303 fps, and YOLOv5s at 238 fps in that order. However, the superiority of YOLO-Jujube over other methods is attributed to its AP performance.

To justify the superiority of YOLO-Jujube, the methods were tested on a test set and the result are displayed in Figure 5. Excluding YOLO-Jujube without missing detection in the blue color circle or false detection in the yellow circle color, one-blue and one-yellow is found in YOLOv3-tiny, one-blue is noted in YOLOv4-tiny, and two-yellow is observed in both YOLOv5s and YOLOv7-tiny. Similarly, the general performance of the methods is tabulated in Table 4. Except for the number of layers, the Params obtained from YOLO-Jujube are 8% higher than YOLOv4-tiny, and 50.4%, 29.5%, 14.3%, respectively, lower than YOLOv3-tiny, YOLOv5s, and YOLOv7-tiny. In the case of GFLOPs, YOLO-Jujube is 9.8%, 24.7%, 29.8%, and 10.5% less than YOLOv3-tiny, YOLOv4-tiny, YOLOv5s, and

YOLOv7-tiny, respectively. This is to justify the training speed of the methods. Meanwhile, the obtained $F_1$ score of YOLO-Jujube, at 84.2%, is 4.4%, 1.1%, 0.1%, and 2.0% greater than YOLOv3-tiny, YOLOv4-tiny, YOLOv5s, and YOLOv7-tiny, respectively. To demonstrate the outstanding performance among the methods, AP is used because it measures the overall relationship of P−R, making it more accurate than the $F_1$-score. Table 4 shows that the AP performance of YOLO-Jujube is outstanding as it is 5.9%, 1.4%, 0.3%, and 3% higher than YOLOv3-tiny, YOLOv4-tiny, YOLOv5s, and YOLOv7-tiny, respectively. Owing to the number of layers, the average detection speed of YOLOv3-tiny at 435 fps is faster than YOLOv4-tiny at 323 fps, YOLO-Jujube at 245 fps, YOLOv7-tiny at 238 fps, and YOLOv5s at 204 fps. The results revealed that the number of layers is a perfect indicator for detection speed correlation compared to Params, which requires further investigation. To establish that the performance of methods is dependent on the conditions of the dataset, our finding using the jujube fruit dataset reveals that regarding the AP and weight-size, respectively, YOLO-Jujube is 8.7% more than and 7.2% less than the proposed method by Zhang et al. [19] for dragon fruit detection. Furthermore, the weight-size is reduced and the detection speed is faster compared to other mainstream YOLO variants such as YOLO-Banana [31], YOLO-Oleifera [14], GCS-YOLOv4-tiny [16], and so on. Considering the performance of all methods, YOLO-Jujube is robust against the complex environment, accurate, fast, and applicable to meet the goal of a computer vision-based understanding of images with ripeness stages.

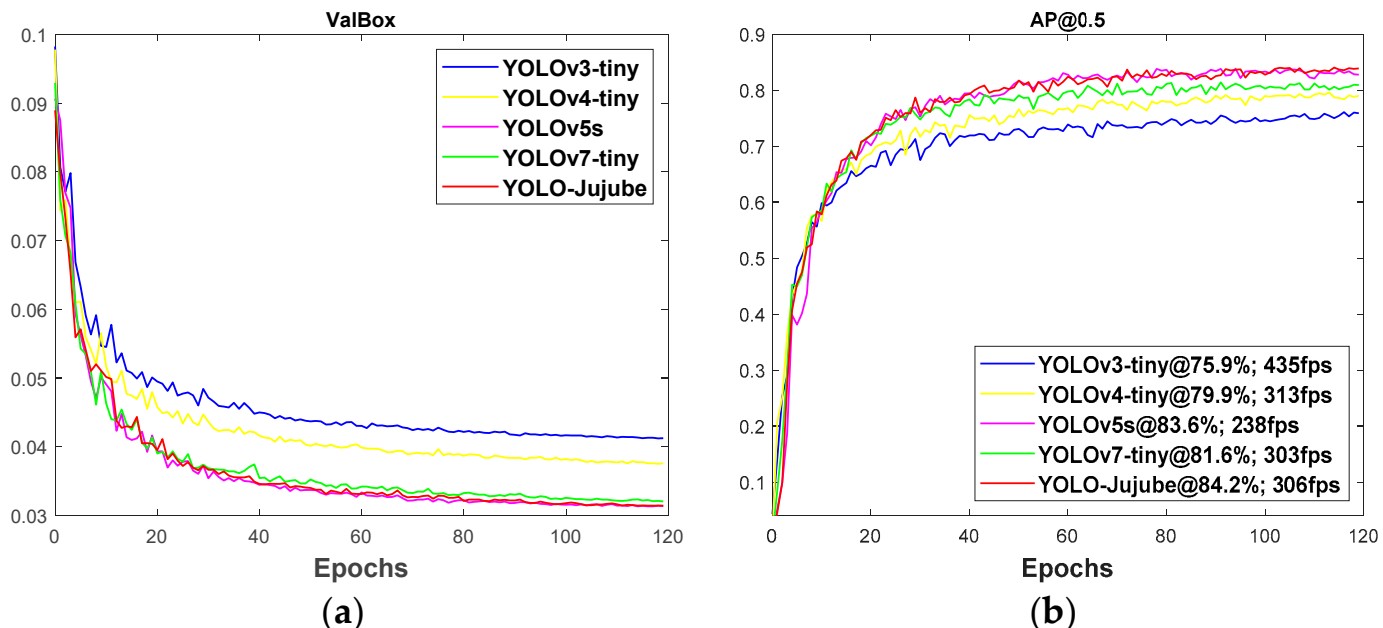

**Figure 4.** Trained methods' results for (**a**) validation box; (**b**) AP@0.5.

**Table 4.** Performance of the methods on the test set.

| Methods | Layers | Param(m) | GFLOPs | P% | R% | $F_1$% | AP% | Avg_fps |
|---|---|---|---|---|---|---|---|---|
| YOLOv3-tiny | 38 | 8.7 | 12.9 | 83.8 | 77.2 | 80.3 | 82.9 | 435 |
| YOLOv4-tiny | 89 | 4.8 | 15.0 | 90.0 | 78.1 | 83.6 | 87.4 | 323 |
| YOLOv5s | 157 | 7.0 | 15.8 | 87.8 | 81.6 | 84.6 | 88.5 | 204 |
| YOLOv7-tiny | 146 | 6.0 | 13.0 | 85.4 | 80.2 | 82.7 | 85.8 | 238 |
| YOLO-Jujube | 134 | 5.2 | 11.7 | 87.9 | 81.7 | 84.7 | 88.8 | 245 |

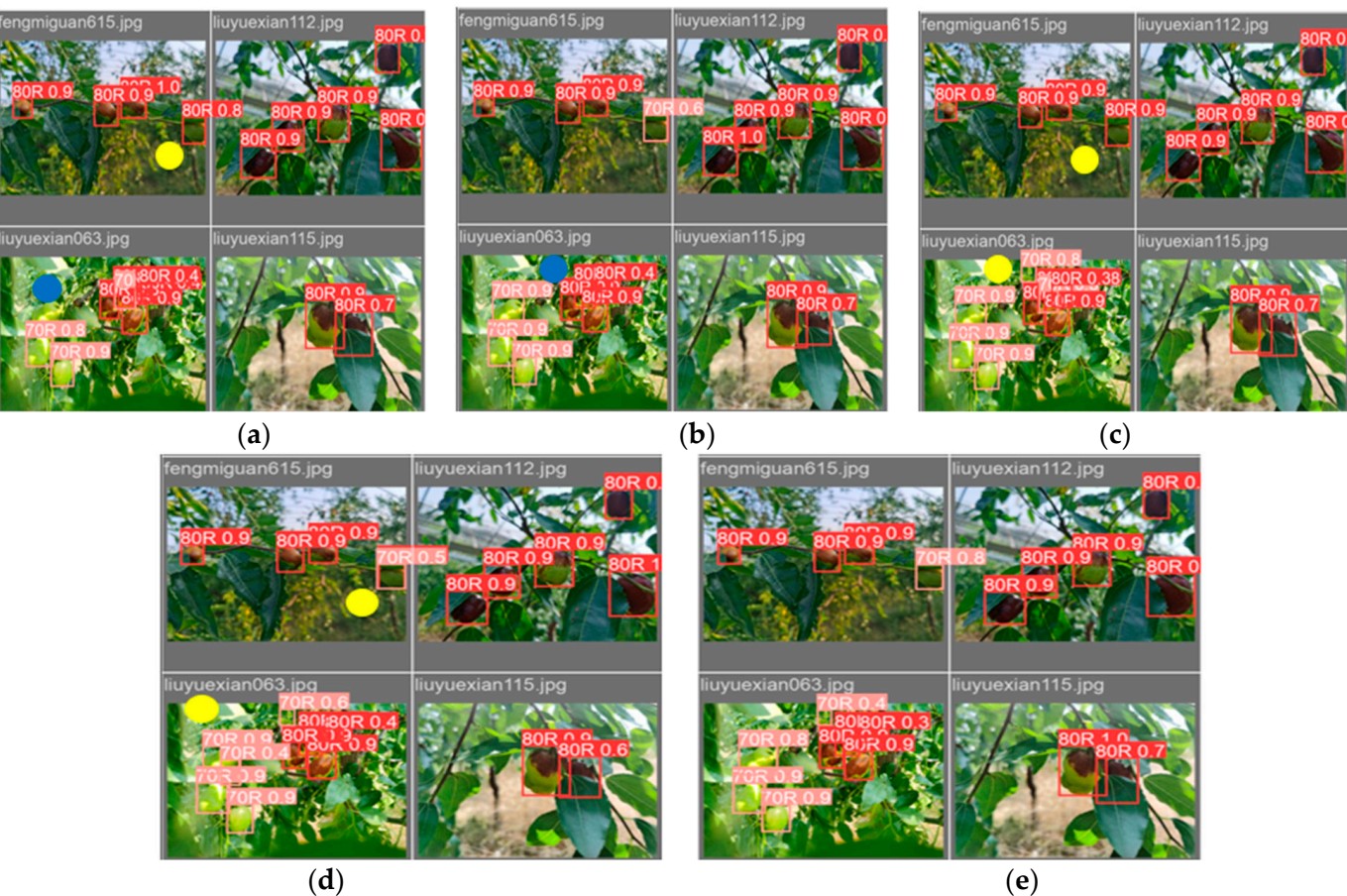

**Figure 5.** Result of four batches of images tested on the test set using the methods: (**a**) YOLOv3-tiny; (**b**) YOLOv4-tiny; (**c**) YOLOv5s; (**d**) YOLOv7-tiny; (**e**) YOLO-Jujube.

### 3.2. Sorting and Counting

The methods were subjected to sorting and counting of target fruits using the test set and videos recorded to examine the level of robustness. The sorting involves the tracking of each ripeness class followed by counting. Under the influence of occlusions and a similar background image, including other complex conditions, the methods could detect a number of target fruits, as displayed in Figures 6 and 7, with their detection speed. YOLO-Jujube, as shown in Figure 6e, is excellent regarding the sorting and counting two targets of 80R with a higher confidence score compared to YOLOv4-tiny (Figure 6b), YOLOv5s (Figure 6c), and YOLOv7-tiny (Figure 6d) with a similar detection. Meanwhile, YOLOv3-tiny (Figure 6a) was observed to have detected four targets, including incorrect detections, showing less robustness. Just like Figure 6, Figure 7 was set for 70R target fruits tracking. YOLO-Jujube, as shown in Figure 7e, sorted and counted seven targets of 70R in the image, with a confidence score higher than YOLOv5s (Figure 7c) with the same detection condition. YOLOv4-tiny (Figure 7b) and YOLOv7-tiny (Figure 7d) sorted and counted eight targets of 70R with incorrect detections, while YOLOv3-tiny (Figure 7a) sorted and counted six targets of 70R with missed detection. This proves that YOLO-Jujube is more robust for sorting and counting compared to other methods, making it the best candidate for fruit ripeness inspection.

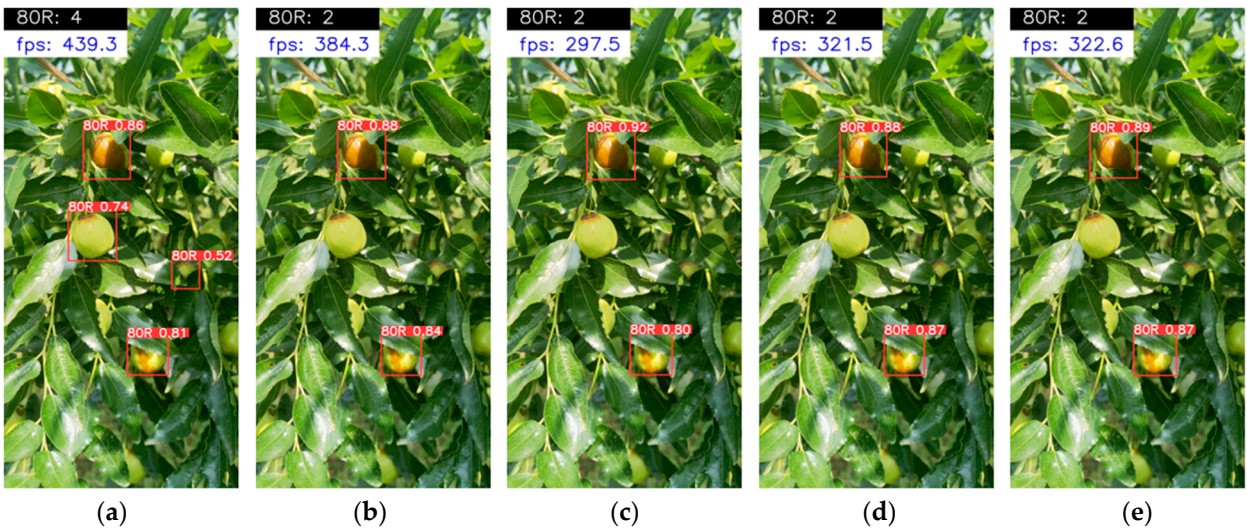

**Figure 6.** The sorted and counted target of 80R from an image tested on: (**a**) YOLOv3-tiny; (**b**) YOLOv4-tiny; (**c**) YOLOv5s; (**d**) YOLOv7-tiny; (**e**) YOLO-Jujube method.

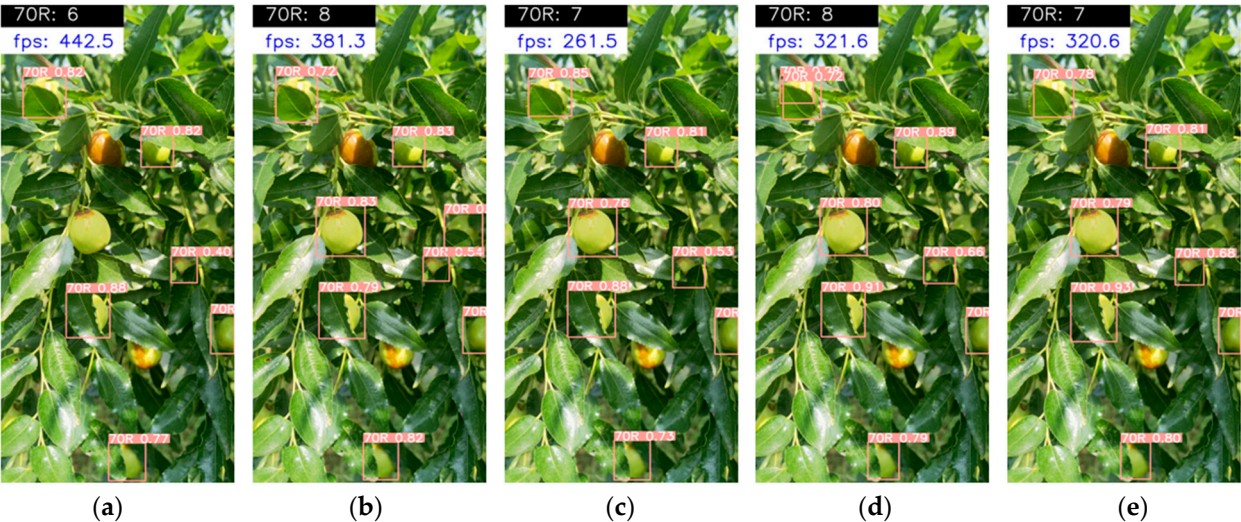

**Figure 7.** The sorted and counted target of 70R from an image tested on: (**a**) YOLOv3-tiny; (**b**) YOLOv4-tiny; (**c**) YOLOv5s; (**d**) YOLOv7-tiny; (**e**) YOLO-Jujube method.

Table 5 shows the sorting and counting summary of target fruits detected in the test set, including their average speed. Meanwhile, the subscript of L, C, and d% represent label, count, and percentage difference, respectively. At an average speed of 357 fps, 286 fps, 227 fps, 256 fps, and 259 fps, respectively, for YOLOv3-tiny, YOLOv4-tiny, YOLOv5s, YOLOv7-tiny, and YOLO-Jujube, the number of sorted and counted targets under $80R_C$ in YOLOv3-tiny is equal to YOLOv4-tiny > YOLOv5s > YOLOv7-tiny > YOLO-Jujube, and the targets under $70R_C$ in YOLOv7-tiny > YOLOv3-tiny > YOLOv4-tiny > YOLO-Jujube > YOLOv5s. Comparing the value obtained from Tables 2–5, an increase in the number of target fruits sorted and counted is indicated by false positive detections. Using Equation (7), the number of false positive detections calculated from $80R_Cd\%$ demonstrates that the 10.72% obtained with YOLO-Jujube is less than 12.35% with YOLOv7-tiny, 14.28% with YOLOv5s, 15.67% with YOLOv4-tiny, and 15.67% with YOLOv3-tiny. In addition, for $70R_Cd\%$, 8.70% with YOLOv5s is less than 9.97% with YOLO-Jujube, 10.60% with YOLOv4-tiny, 17.25% with YOLOv3-tiny, and 27.19% with YOLOv7-tiny. The overall outcome reveals that the number of false positives detected on YOLO-Jujube is lower, confirming the robustness compared to other methods.

**Table 5.** Summary of sorting and counting on images in the test set.

| Methods | Avg_fps | $80R_L$ | $80R_C$ | $70R_L$ | $70R_C$ | $80R_Cd\%$ | $70R_Cd\%$ |
|---|---|---|---|---|---|---|---|
| YOLOv3-tiny | 357 | 97 | 578 | 53 | 170 | 15.67 | 17.25 |
| YOLOv4-tiny | 286 | 96 | 578 | 54 | 159 | 15.67 | 10.60 |
| YOLOv5s | 227 | 96 | 570 | 52 | 156 | 14.29 | 8.70 |
| YOLOv7-tiny | 256 | 96 | 559 | 56 | 188 | 12.35 | 27.19 |
| YOLO-Jujube | 259 | 96 | 550 | 53 | 158 | 10.72 | 9.97 |

Having observed a similarity in the results on tested videos, real-time detection tested on a video of 199.2 MB confirmed that the obtained average speed is similar in trend to the outcome from Tables 4 and 5. Table 6 indicates the average detection speed of the compared methods, including the total detected targets counted from the total number of frames in the video. The number of sorted and counted for $80R_C$ is the same for all methods, but with different variations in $70R_C$, whereas the number of fruits target detections noted in YOLO-Jujube has 256 more than the other method with the most detections, which is YOLOv3-tiny. The re-write video of each method remains the aspect of a future study because the number of detected fruits targeted do not correlate with obtained weight of the video for both $80R_CV_S$ and $70R_CV_S$. For YOLO-Jujube, having the re-write video of 585.8 MB under $80R_CV_S$ is higher than other methods, but becomes the least at 521.6 MB under $70R_CV_S$. Nonetheless, as $80R_C$ and $70R_C$ are created from $80R_L$ and $70R_L$ respectively, the number of labels generated under YOLO-Jujube from $80R_L$ and $70R_L$ is, respectively, 2303 and 1511, which is more than other methods.

**Table 6.** Re-write video: Sorting and counting on 199.2MB video.

| Methods | Avg_fps | $80R_L$ | $80R_C$ | $70R_L$ | $70R_C$ | $80R_CV_S$ (MB) | $70R_CV_S$ (MB) |
|---|---|---|---|---|---|---|---|
| YOLOv3-tiny | 370 | 2271 | 9998 | 1405 | 3841 | 585.2 | 526.7 |
| YOLOv4-tiny | 303 | 2279 | 9998 | 1479 | 3246 | 576.5 | 526.3 |
| YOLOv5s | 244 | 2272 | 9998 | 1503 | 3600 | 576.7 | 524.2 |
| YOLOv7-tiny | 270 | 2312 | 9998 | 1308 | 3199 | 578.8 | 527.0 |
| YOLO-Jujube | 276 | 2303 | 9998 | 1511 | 4097 | 585.8 | 521.6 |

## 4. Conclusions

The ability to detect the ripeness stages of jujube fruits is essential for inspection of an automatic picking method. However, fruit detection is confronted with the challenges of a complex and changing environment, including the phases of jujube ripeness. For this reason, a YOLO-Jujube method was proposed in this paper based on YOLOv5 architecture and validated using a new jujube fruits image dataset. The YOLO-Jujube incorporated a network of Stem, RCC, Maxpool, CBS, SPPF, C3, PANet, and CIoU loss to improve the fruit detection performance. The performance demonstrated that the Params from YOLO-Jujube are 8% higher than YOLOv4-tiny, and 50.4%, 29.5%, and 14.3%, respectively, lower than YOLOv3-tiny, YOLOv5s, and YOLOv7-tiny. For GFLOPs, to justify the training speed, YOLO-Jujube is 9.8%, 24.7%, 29.8%, and 10.5% less than YOLOv3-tiny, YOLOv4-tiny, YOLOv5s, and YOLOv7-tiny, respectively. In the case of AP, YOLO-Jujube is outstanding as it is 5.9%, 1.4%, 0.3%, and 3% higher than YOLOv3-tiny, YOLOv4-tiny, YOLOv5s, and YOLOv7-tiny, respectively. At the same time, the detection speed, which is dependent on layers, is measured as follows: YOLOv3-tiny 435 fps, YOLOv4-tiny 323 fps, YOLO-Jujube 245 fps, YOLOv7-tiny 238 fps, and YOLOv5s 204 fps. The calculated number of false positives detected during the sorting and counting process found in YOLO-Jujube is less than YOLOv3-tiny, YOLOv4-tiny, YOLOv5s, and YOLOv7-tiny. In all, YOLO-Jujube is robust and applicable for ripeness inspection. Future work will further improve the detection performance of the proposed method in terms of accuracy and a smaller number of network parameters, and to make it extendable to other agricultural products for investigation.

**Author Contributions:** Conceptualization, D.X. and H.Z.; methodology, H.Z. and O.M.L.; validation, D.X., H.Z. and S.Z.; formal analysis, H.Z.; resources, D.X., H.Z., X.L. and R.R.; data curation, D.X., H.Z., X.L. and R.R.; writing—original draft preparation, O.M.L.; writing—review and editing, D.X. and H.Z.; project administration, D.X.; funding acquisition, D.X. and H.Z. All authors have read and agreed to the published version of the manuscript.

**Funding:** This research was funded by the Key R&D project of introducing high-level scientific and technological talents in Lvliang City, grant number 2021RC-2-24, and the Basic Research Project of Shanxi Province, grant number 202103021223145.

**Data Availability Statement:** The data used to support the findings of this study are available from the corresponding author upon request.

**Conflicts of Interest:** The authors declare no conflict of interest.

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
