# Peer review of "An Automatic Jujube Fruit Detection and Ripeness Inspection Method in the Natural Environment"

_agronomy, doi:10.3390/agronomy13020451_

Round 1
Reviewer 1 Report
General Comments:
1. This manuscript provides development and evaluation of fruit detection and Ripeness inspection with object detection model in natural field condition.
- Language of the manuscript needs considerable edit. It is suggested that the authors may consult a professional English editor for improvement.
- Jujube detection with object detection models (YOlO4-tiny, YOLO5s, YOLOv7-tiny, and YOLO-Jujube) identifies jujube separately in each input frame of video, therefore the same jujube can be counted several times if a previously recognized and counted jujube appears in the next frame.
Title:
“Mothod in Natural Enviroment” should be Method in Natural Environment
Abstract:
Line 16: on which platform the model was deployed? This should be mentioned because fps depends on the computing power and size of the object recognition model.
Materials and Methods:
Please describe how the authors counted jujube only with the object detection model in the collected video.
Results:
Figures 6 and 7 could be merged to show the amount of 80R and 70R in a single image, as well as the overall amount of detected Jujube fruit.
Author Response
1、This manuscript provides development and evaluation of fruit detection and Ripeness inspection with object detection model in natural field condition.
Response: Thanks for your support.
2、Language of the manuscript needs considerable edit. It is suggested that the authors may consult a professional English editor for improvement.
Response: Thank you. Noted and manuscript updated.
3、Jujube detection with object detection models (YOlO4-tiny, YOLO5s, YOLOv7-tiny, and YOLO-Jujube) identifies jujube separately in each input frame of video, therefore the same jujube can be counted several times if a previously recognized and counted jujube appears in the next frame.
Response: Thank you. Yes, the same jujube can be counted several times. However, the targets detected were counted on separated frame of video and the total detected targets counted from the total number of frames.
4、Title:
“Mothod in Natural Enviroment” should be Method in Natural Environment
Response: Thanks for your support. Noted and corrected in the updated manuscript.
5、Abstract:
Line 16: on which platform the model was deployed? This should be mentioned because fps depends on the computing power and size of the object recognition model.
Response: Thanks for your contributions. The model was deployed on computer as detailed in section 2.3
6、Materials and Methods:
Please describe how the authors counted jujube only with the object detection model in the collected video.
Response: Thanks. Noted and included in the updated manuscript. The number of frames in a video was determined and jujube targets counted separately in each frame.
7、Results:
Figures 6 and 7 could be merged to show the amount of 80R and 70R in a single image, as well as the overall amount of detected Jujube fruit.
Response: Thanks for your suggestions. Merging Figures 6 and 7 will affect the aim of sorting and counting designed in the paper.

Reviewer 2 Report
An automatic Jujube Fruit Detection and Ripeness Inspection Mothod in Natural Enviroment
The paper is a good example of application of artificial intelligence in a natural environment
One valuable aspect of the work is the fact that images are taken in open field with two different cameras "at white and crisp mature stage, ... in the morning, midday and afternoon with constantly changing shooting angle and distance. The captured images include complex and changeable conditions such as varying maturity, leaf occlusion, fruit overlap occlusion, superimposed fruit, dense target fruit, branch occlusion, back light, front light, side light and other fruit scenes."
Such variability is an importan part of the work: please add a table indicating the total numbers for each group (dates, time of the day, ripening stage, distance and overlp/occlusion)
I do not understand why the analyses were divided into 70R and 80R: did you have both in the same image? or it is just on the basis of different dates? if it is a divisioon just made on the basais of different dates, I don't see a need to separate results. If both might be present in the same image, I do not understand how less or more mature fruits occasionally present in the images have been treated.
Also, as visible in figure 1d, there are also some bruised fruits: how have such fruits (with defects) been treated? please discuss in the paper
Please specify the meaning of 80RL 80RC 70RL 70RC 80RCd% 70RCd% in table 4.
I do not see the reason why a new "YOLO-Jujube" has been developed is it a method that worrks exclusivekly for jujube? is it substantially different from other YOLO? If it is not substantially different (as in my impression) there is no need to give a new name to something that is just an adaptation to a specific casae study.
As often happens the test has been done in a sub-group of the initial sample. It would have been interesting and would be more robust to apply the method also to a completely different jujube dataset (images can be found freely available in the web)
I do not fully understand: the method has been applied to 1959 images and additionally to a video? or were images also extracted from the video? why considering only one video?
the paper has been proposed for publication in Agronomy: the work is indeed fitting the scopes of the journal, hwever authors have not considered previous work published by the same journal. see e.g.: https://www.mdpi.com/search?q=yolo&journal=agronomy
Author Response
1、An automatic Jujube Fruit Detection and Ripeness Inspection Mothod in Natural Enviroment
Response: Thank you. Noted and corrected.
2、The paper is a good example of application of artificial intelligence in a natural environment
Response: Thanks for your support.
3、One valuable aspect of the work is the fact that images are taken in open field with two different cameras "at white and crisp mature stage, ... in the morning, midday and afternoon with constantly changing shooting angle and distance. The captured images include complex and changeable conditions such as varying maturity, leaf occlusion, fruit overlap occlusion, superimposed fruit, dense target fruit, branch occlusion, back light, front light, side light and other fruit scenes."
Response: Thanks for your contributions.
4、Such variability is an important part of the work: please add a table indicating the total numbers for each group (dates, time of the day, ripening stage, distance and overlp/occlusion)
Response: Thanks for your suggestions. Figure 1 shows some of the obtained images in the morning, mid-day and afternoon under different conditions.
5、I do not understand why the analyses were divided into 70R and 80R: did you have both in the same image? or it is just on the basis of different dates? if it is a divisioon just made on the basais of different dates, I don't see a need to separate results. If both might be present in the same image, I do not understand how less or more mature fruits occasionally present in the images have been treated.
Response: Thanks for your contributions and suggestions. The analyses were divided into 80R (Crisp mature) and 70R (White mature) for ripeness inspection towards sorting and counting. Yes, have both in the same image or as separate targets in image. This is to investigate the level of jujube ripeness.
6、Also, as visible in figure 1d, there are also some bruised fruits: how have such fruits (with defects) been treated? please discuss in the paper
Response: Thanks. No defective images were taken. The paper focuses on ripeness inspection using a fruit detection method. However, the incorporation of bruised fruits would require a new dataset.
7、Please specify the meaning of 80RL 80RC 70RL 70RC 80RCd% 70RCd% in table 4.
Response: Thanks for your contributions. Noted and manuscript updated. The subscript L, C and d% is label, count and percentage difference, respectively in Table 4.
8、I do not see the reason why a new "YOLO-Jujube" has been developed is it a method that worrks exclusivekly for jujube? is it substantially different from other YOLO? If it is not substantially different (as in my impression) there is no need to give a new name to something that is just an adaptation to a specific casae study.
Response: Thanks for your contributions. YOLO-Jujube is simply an improved YOLOv5 network. YOLO-Jujube provides a unique name for the method. However, it is applicable to other fruit detection networks.
9、As often happens the test has been done in a sub-group of the initial sample. It would have been interesting and would be more robust to apply the method also to a completely different jujube dataset (images can be found freely available in the web)
Response: Thanks for your suggestions. The test-set provided in Table 1 including video, is completely a different dataset used to investigate the robustness of YOLO-Jujube. The annotation of the test-set was only provided to experiment sorting and counting processes.
10、I do not fully understand: the method has been applied to 1959 images and additionally to a video? or were images also extracted from the video? why considering only one video?
Response: Thank you. Table 1 provides the dataset details. Several videos of Jujube fruits were investigated, not only one. But the result of one video was presented in the paper.
11、the paper has been proposed for publication in Agronomy: the work is indeed fitting the scopes of the journal, hwever authors have not considered previous work published by the same journal. see e.g.: https://www.mdpi.com/search?q=yolo&journal=agronomy
Response: Thanks for your contributions and suggestions. Reference 31 added.

Reviewer 3 Report
The authors propose YOLO-Jujube, a method capable of detecting jujube fruit automatically with better performance in comparison to other YOLO variants. The work is very interesting as it is applied to a practical issue, to increase production. In order to make the paper more clear to readers, some points should be improved, as listed as follows.
Authors state that "... which helps to replace the traditional hand-picking principle.". How is the proposed algorithm replacing the traditional hand-picking approach? The proposed solution may be used to help workers easily find the Jujube fruit, but the hand-picking continues to happen. The authors should present some automated solution for harvesting the fruits given their identification or, if its the case, just modify the sentence to state that the proposed system can be part of an automated solution.
Sometimes the authors repeat the same information that is already listed in the table, for example: "Table 5 indicates that the average detection speed of YOLOv3-tiny is 370 fps, YOLOv4-tiny is 303 fps, 308 YOLO-Jujube is 276 fps, YOLOv7-tiny is 270 fps and YOLOv5s is 244 fps." This is not necessary. Please avoid this duplicity in all occurrences on the text (this mainly occurs with data shown in tables).
Authors must justify more clearly why they are only using "tiny" variants, of the YOLO algorithm. The "embedded solution" or "edge processing" should be highlighted in order to show the importance of such approach. This will add to the value of the work.
Some comparisons made by the authors need improvement, as they are not clear. For instance, "where the number of fruits target detection noted in YOLO-Jujube is 4097 higher than other methods." This is not correct, the value is not relative to the other methods, so YOLO-Jujube does not have 4097 more detections than the other methods. You may say that YOLO-Jujube has 256 more detections than the method with the most detections (which is YOLOvv3-tiny, wich 3841).
Some general comments and writing errors are listed as follows.
"Mothod" -> "Method"
"the sweeter is the fruits." -> "the sweeter is the fruit."
"It has widely utilized" -> "It has been widely utilized"
"contains of computer vision" -> "contains computer vision"
"YOLOv3-tiny reported" -> "YOLOv3-tiny, reported"
"29.4 frame per" -> "29.4 frames per"
"[16]model" -> "[16] model"
"in dense small image" -> "in dense small images"
"weight-size still remain large."-> "weight-size still remains large."
"YOLOv4 was presented." -> "YOLOv4, was presented."
"[19]introduced" -> "[19] introduced"
"particularly jujube fruits" -> "particular to jujube fruits"
"have surpasses" -> "have surpassed"
"A comparatively large weight-size and parameters common to all the said YOLO networks, which is a big challenge to realize a faster detection speed." -> please rewrite
"small and dense image." -> "small and dense images."
"number of C3 network in Figure 2 was pruned one" -> "number of C3 networks in Figure 2 was pruned to one"
"2.3 Experiment" -> adjust font
"performance was precision" -> "performance were precision"
"Eq. 6 respectively." -> "Eq. 6, respectively."
"in frame per second" -> "in frames per second"
"It is measure as" -> "It is measured as"
"numbers. Where" -> "numbers, where"
"Figureure 4" -> "Figure 4"
"YOLOv7-tiny respectively."-> "YOLOv7-tiny, respectively."
"respectively. Owing" -> please remove extra space
"This shows the number of layers to be more coherent than Params as a perfect indicator for detection speed requires further investigation." -> please rewrite
"[19]for" -> "[19] for"
Figure 6 is shifted to the right.
"target of 80R" -> "targets of 80R"
"target of 70R" -> "targets of 70R"
"counted target under 80RC" -> "counted targets under 80RC"
"the target under" -> "the targets under"
"is measure as" -> "is measured as"
Author Response
The authors propose YOLO-Jujube, a method capable of detecting jujube fruit automatically with better performance in comparison to other YOLO variants. The work is very interesting as it is applied to a practical issue, to increase production. In order to make the paper more clear to readers, some points should be improved, as listed as follows.
Response: Thanks for your support and contributions.
1、Authors state that "... which helps to replace the traditional hand-picking principle.". How is the proposed algorithm replacing the traditional hand-picking approach? The proposed solution may be used to help workers easily find the Jujube fruit, but the hand-picking continues to happen. The authors should present some automated solution for harvesting the fruits given their identification or, if its the case, just modify the sentence to state that the proposed system can be part of an automated solution.
Response: Thanks. Noted and corrected in the updated manuscript.
2、Sometimes the authors repeat the same information that is already listed in the table, for example: "Table 5 indicates that the average detection speed of YOLOv3-tiny is 370 fps, YOLOv4-tiny is 303 fps, 308 YOLO-Jujube is 276 fps, YOLOv7-tiny is 270 fps and YOLOv5s is 244 fps." This is not necessary. Please avoid this duplicity in all occurrences on the text (this mainly occurs with data shown in tables).
Response: Thanks for your support. Noted and manuscript updated.
3、Authors must justify more clearly why they are only using "tiny" variants, of the YOLO algorithm. The "embedded solution" or "edge processing" should be highlighted in order to show the importance of such approach. This will add to the value of the work.
Response: Thanks for your support. Noted and manuscript updated.
4、Some comparisons made by the authors need improvement, as they are not clear. For instance, "where the number of fruits target detection noted in YOLO-Jujube is 4097 higher than other methods." This is not correct, the value is not relative to the other methods, so YOLO-Jujube does not have 4097 more detections than the other methods. You may say that YOLO-Jujube has 256 more detections than the method with the most detections (which is YOLOvv3-tiny, wich 3841).
Response: Thanks for your support. Noted and corrected.
5、Some general comments and writing errors are listed as follows.
"Mothod" -> "Method"
"the sweeter is the fruits." -> "the sweeter is the fruit."
"It has widely utilized" -> "It has been widely utilized"
"contains of computer vision" -> "contains computer vision"
"YOLOv3-tiny reported" -> "YOLOv3-tiny, reported"
"29.4 frame per" -> "29.4 frames per"
"[16]model" -> "[16] model"
"in dense small image" -> "in dense small images"
"weight-size still remain large."-> "weight-size still remains large."
"YOLOv4 was presented." -> "YOLOv4, was presented."
"[19]introduced" -> "[19] introduced"
"particularly jujube fruits" -> "particular to jujube fruits"
"have surpasses" -> "have surpassed"
"A comparatively large weight-size and parameters common to all the said YOLO networks, which is a big challenge to realize a faster detection speed." -> please rewrite
"small and dense image." -> "small and dense images."
"number of C3 network in Figure 2 was pruned one" -> "number of C3 networks in Figure 2 was pruned to one"
"2.3 Experiment" -> adjust font
"performance was precision" -> "performance were precision"
"Eq. 6 respectively." -> "Eq. 6, respectively."
"in frame per second" -> "in frames per second"
"It is measure as" -> "It is measured as"
"numbers. Where" -> "numbers, where"
"Figureure 4" -> "Figure 4"
"YOLOv7-tiny respectively."-> "YOLOv7-tiny, respectively."
"respectively. Owing" -> please remove extra space
"This shows the number of layers to be more coherent than Params as a perfect indicator for detection speed requires further investigation." -> please rewrite
"[19]for" -> "[19] for"
Figure 6 is shifted to the right.
"target of 80R" -> "targets of 80R"
"target of 70R" -> "targets of 70R"
"counted target under 80RC" -> "counted targets under 80RC"
"the target under" -> "the targets under"
"is measure as" -> "is measured as"
Response: Thanks so much. All noted and corrected in the updated manuscript.

Reviewer 4 Report
The paper presented a task of detecting fruits, specifically jujubes, is difficult due to the dynamic nature of the environment and different stages of ripeness. To address this, the authors propose using YOLO-jujubes architecture and evaluate it using a newly created dataset of jujube images. The methodology part presents the applied approach. The results are supported with figures and statistical outputs. The obtained results could be beneficial for similar studies; however, the authors should consider the questions and suggestions written below;
1. The authors are invited to make Space before each reference
2. Thus, change YOLO-Jujube to YOLO in keywords
3. In line 35, add reference in the last paragraph.
4. align all tables and figures with their titles
5. We would like to see the original structure of yolov7 for example in ordre to compare it with your proposed architecture.
6. We suggest adding the link of the dataset for giving an opportunity to the scientific community to access the data easily.
7. As well as the authors are invited to avoid large space after line 260
8. Thus, in line 271, change color of speed word
9. Justify paragraph after the line 306
10. In Table 5 , pay attention to the table title : 80RCVS(MB) and 70RCVS(MB)
11. It is necessary to add a detailed explanation about sorting and counting.
12. Are authors used the images include complex and changeable conditions in experimental study? If yes, they need to add some results for each conditions.
13. In conclusion, more discussions should be made regarding the limitation and future dimensions of the proposed study
Author Response
The paper presented a task of detecting fruits, specifically jujubes, is difficult due to the dynamic nature of the environment and different stages of ripeness. To address this, the authors propose using YOLO-jujubes architecture and evaluate it using a newly created dataset of jujube images. The methodology part presents the applied approach. The results are supported with figures and statistical outputs. The obtained results could be beneficial for similar studies; however, the authors should consider the questions and suggestions written below;
Response: Thanks for your support.
1、The authors are invited to make Space before each reference
Response: Thanks for your support and contributions.
2、Thus, change YOLO-Jujube to YOLO in keywords
Response: Thanks for your suggestions. However, YOLO-Jujube is used to for the improved YOLOv5.
3、In line 35, add reference in the last paragraph.
Response: Thanks for your suggestions. Noted. This is just general knowledge about the traditional hand-picking method being replaced by an automatic method.
4、align all tables and figures with their titles
Response: Thanks. Noted and corrected in the updated manuscript.
5、We would like to see the original structure of yolov7 for example in ordre to compare it with your proposed architecture.
Response: Thanks for your suggestions. YOLOv7 [24] is referenced and compared to YOLO-Jujube in the paper. But our proposed YOLO-Jujube was based on improved YOLOv5 network.
6、We suggest adding the link of the dataset for giving an opportunity to the scientific community to access the data easily.
Response: Thanks. Noted.
7、As well as the authors are invited to avoid large space after line 260
Response: Thanks for your support and contributions. Corrected in the updated manuscript.
8、Thus, in line 271, change color of speed word
Response: Thank you. Noted and corrected.
9、Justify paragraph after the line 306
Response: Thanks for your contributions. Noted
10、In Table 5 , pay attention to the table title : 80RCVS(MB) and 70RCVS(MB)
Response: Thanks for your support. Noted. 80RCVS(MB) and 70RCVS(MB) are simply the re-write video generated using the compared methods.
11、It is necessary to add a detailed explanation about sorting and counting.
Response: Thanks for your contributions. The sorting and counting have been detailed in section 3.2. “The sorting involves the tracking of each ripeness class followed by counting”
12、Are authors used the images include complex and changeable conditions in experimental study? If yes, they need to add some results for each conditions.
Response: Thanks for your suggestions. Figure 1 shows some of the obtained images under different conditions. However, the summary of the detection performance was presented in the paper. Using the entire conditions will involve more incorporation of sorting and counting.
13、In conclusion, more discussions should be made regarding the limitation and future dimensions of the proposed study
Response: Thanks for your suggestions. Noted.

Round 2
Reviewer 2 Report
The paper has been only partially improved.
In my previous review I wrote:
The captured images include complex and changeable conditions such as varying maturity, leaf occlusion, fruit overlap occlusion, superimposed fruit, dense target fruit, branch occlusion, back light, front light, side light and other fruit scenes." Such variability is an important part of the work: please add a table indicating the total numbers for each group (dates, time of the day, ripening stage, distance and overlp/occlusion)
Response: Thanks for your suggestions. Figure 1 shows some of the obtained images in the morning, mid-day and afternoon under different conditions.
Referee: The authors thanked for the suggestion but didn't actually add a table as recommended. Please add as recommended.
10、I do not fully understand: the method has been applied to 1959 images and additionally to a video? or were images also extracted from the video? why considering only one video?
Response: Thank you. Table 1 provides the dataset details. Several videos of Jujube fruits were investigated, not only one. But the result of one video was presented in the paper.
Referee: Table 1 does not include videos: please explain. Why considering only one video? please explain
Round 2
# Reviewer 2
1、Referee: The authors thanked for the suggestion but didn't actually add a table as recommended. Pleaseadd as recommended.
Response: Thanks for your support. Noted. The table added as recommended.
2、Referee: Table 1 does not include videos: please explain, Why considering only one video? pleaseexplain
Response: Thanks for your contributions. A total of 1959 images were taken excluding videos. Yes, the table does not include video because it is not annotated, taken as unseen data subjected to open countingof fruit targets. However, the video is part of the obtained dataset. Actually, three videos were recorded but only one video was presented because we noticed that the results obtained on tested methods were approximately the same.
